# Stray Light Nonuniform Background Elimination Method Based on Image Block Self-Adaptive Gray-Scale Morphology for Wide-Field Surveillance

**Jianing Wang** [1,2]**, Xiaodong Wang** [1,]*** and Yunhui Li** [1]

[1] Changchun Institute of Optics, Fine Mechanics and Physics, Chinese Academy of Sciences, Changchun 130033, China; wangjianing17@mails.ucas.ac.cn (J.W.); liyunhui@ciomp.ac.cn (Y.L.)
[2] University of Chinese Academy of Sciences, Beijing 100049, China
* Correspondence: wangxd@ciomp.ac.cn

**Abstract:** Space-based wide-field surveillance systems are of great significance in maintaining the security of space resources by avoiding collisions between space targets. However, their performance is hindered by stray light phenomena. The nonuniform background noise caused by stray light significantly hampers subsequent target detection, leading to a high frequency of false alarms. To solve this problem, we propose a robust and accurate nonuniform background elimination method based on image block self-adaptive gray-scale morphology (IBSGM). First, we define two kinds of structural operators with different sizes and domains, which make full use of the difference between the target pixels and surrounding background pixels. Then, we block the original surveillance image and find the size of the largest target in each block by the minimum bounding rectangle method to determine the optimal size of the structural operator suitable for each block. Finally, we perform morphological processing using the defined structural operators to eliminate nonuniform backgrounds from images. Experimental results on simulated and real image datasets demonstrate that the proposed IBSGM method has higher precision in eliminating the nonuniform background when compared to other methods.

**Keywords:** stray light; wide-field surveillance; minimum bounding rectangle; gray-scale morphology; background elimination

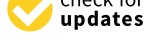



## 1. Introduction

Since Sputnik-1 was launched in 1957, the pace of space exploration quickened [1,2], and the number of targets in space (including satellites and space debris) has increased rapidly [3,4]. According to the Union of Concerned Scientists' satellite database, as of 1 January 2022, there were 4852 satellites in orbit accompanied by a mass of space debris. The famous scientist Kessler pointed out that once these targets collide, every collision could trigger a cascade of collisions that create more space debris and pose a massive threat to operational satellites [5]. For example, space debris of about 10 cm in diameter can destroy any operational satellite it collides with, which would have severe implications for human space activities [6]. To address the growing demand for space exploration while also assuring the security of the space environment, countries all over the world are installing a strategic layout of space-based surveillance systems; for example, the Fengyun satellite program of China [7], the Space Surveillance System of Canada (CSSS), the Space-Based Space Surveillance system (SBSS) [8,9], the Space Tracking and Surveillance System (STSS) and the Space-Based Infrared System (SBIRS) [10] of the USA. However, as space-based surveillance systems operate in outer space, most of the detected space targets are dim with high magnitude. Accordingly, stray light can have a severe impact on the detection performance of space-based surveillance systems [11]. In such cases, the surveillance image is the only data source. If stray light cannot be effectively suppressed, it will create a harsh

nonuniform background signal in the surveillance image, reducing its clarity and dynamic range. Furthermore, substantial degradation effects can appear in surveillance images that greatly obstruct the subsequent recognition and detection of space targets [12,13]. In conclusion, stray light noise badly affects the normal operation and detection ability of these systems. In addition, a surveillance image may represent a vast number of targets of various sizes [14]. Precisely eliminating nonuniform backgrounds generated by stray light without losing targets of various sizes is a challenge that remains. As a result, research on nonuniform background elimination methods for surveillance images is urgently needed.

In recent years, the reduction of stray light has become a problem not only in the field of instrument measurement, but also in the preprocessing of faint target detection and tracking. At present, current nonuniform background elimination methods for surveillance images can be roughly divided into two types: a (1) parametric model-based method, and an (2) image feature-based method. In parametric model-based methods, the nonuniform background is eliminated by constructing a parametric model. Specifically, abundant images are collected in advance, then classified to create a training data set to build a parametric model, such as a spline function or polynomial model, or recent deep learning-based models [15,16]. However, these algorithms are not very effective in eliminating nonuniform backgrounds from surveillance images. On the one hand, parametric models must be accurate, such as the order or the number of terms in the polynomial. However, in practice, there are huge differences in the distributions of nonuniform backgrounds in different images, which makes it difficult to accurately determine the parameters under different conditions or scenes. In addition, even if we have enough surveillance images for training a parametric model, there may still be some unknown cases that make it difficult for fixed-parameter models to eliminate nonuniform backgrounds from surveillance images that are not included in the dataset. On the other hand, some targets with low signal-to-noise ratios (SNRs) exist in surveillance images. Minor flaws in the parametric model development process could result in target losses or greater false alarm rates, lowering the accuracy of subsequent target recognition [17].

With image feature-based methods, the problems of nonuniform background elimination can be solved well in unknown situations because only the imaging features of the image itself need to be considered. These methods can be divided into frequency domain and spatial domain methods. For the frequency domain method, wavelet-based and curvelet-based approaches are most common [18,19]. Due to the demand for domain conversions, these methods are too complex and require much computation time. We need to utilize the difference between the target and background noise in the frequency domain to eliminate the background; however, they are very similar in the frequency domain, which can cause confusion. Since it is difficult to distinguish a low-SNR target from a nonuniform background [20], the accuracy to eliminate the nonuniform background is limited. For the spatial–domain method, since the image is processed directly in the spatial domain, it tends to have a faster runtime than frequency–domain filtering. Methods include mean iterative filtering [21], filtering based on average or gradient thresholding [22], new star target segmentation (NSTS) [23], morphology operation [24] and improved new top-hat transformation (INTHT) [25]. Although these spatial–domain filtering methods can be used to eliminate nonuniform backgrounds from surveillance images, they have some disadvantages: (1) the calculation includes all pixels in the filter, which will have a significant impact on the accuracy of nonuniform background estimation, and (2) they are quite sensitive to the size of the filter. An unreasonable size will decrease the background elimination effect and could even cause some targets to be lost. Therefore, effective and reliable elimination of nonuniform backgrounds from surveillance images while retaining targets remains an open challenge.

To solve these problems, a new, accurate and robust stray light nonuniform background elimination method is proposed, named image block self-adaptive gray-scale morphology (IBSGM). A flowchart of the procedure is shown in Figure 1, which can be broken down into three steps: (1) definition of structural operators, (2) division of the

original surveillance image into blocks and determination of the optimal size of the structural operator suitable for each block by the method of the minimum bounding rectangle, and (3) a morphological operation based on constructed structural operators to estimate and eliminate the nonuniform background. In the first step, two structural operators with varying sizes and domains are defined. We analyze the features of surveillance images and the gray value difference between the target pixels and surrounding background pixels, which provides evidence for constructing new structural operators. In the second step, we block the original surveillance image and find the size of the largest target in each image block by the method of the minimum bounding rectangle. This establishes the optimal size of structural operator suitable for each image block. In the third step, we perform morphological processing on the surveillance images using the designed structural operators. With structural operators of the optimal size, we can reliably conserve target pixels while only using pixels from the surrounding background in the morphological operations. In this way, we solve the problem of surveillance images being sensitive to the size of the filter, which plagued previous methods. Finally, experiment with simulated image datasets and real acquired image datasets show that the proposed ISSGM approach eliminates stray light nonuniform backgrounds with greater accuracy and robustness than other methods.

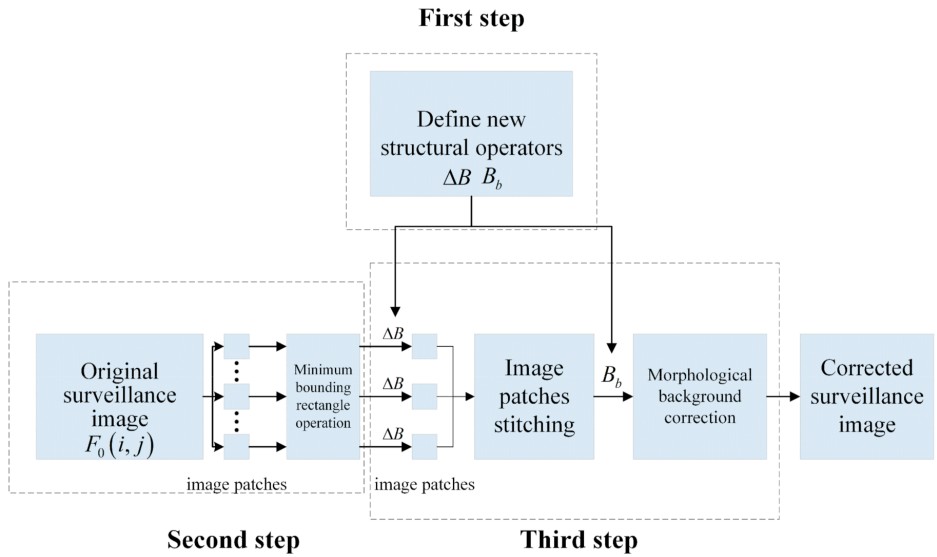

**Figure 1.** Flowchart of the proposed IBSGM method.

## 2. Principles of the Formation and Elimination of Stray Light Backgrounds

Stray light is non-imaging light that radiates to the detector surface or imaging light that propagates via an abnormal path and reaches the detector. The influence of the formation mechanism and the principle of eliminating nonuniform backgrounds from surveillance images will be discussed in detail in this section. There are different sources of stray light, which are of three main types: (1) internal radiation stray light, which is the infrared heat radiation generated by high-temperature optomechanical components such as control motors and temperature-controlled optics inside the system during its normal operation; (2) non-target stray light beyond the field of view, which refers to non-target optical signals that propagate directly or indirectly on the focal plane of the image sensor, and which come from radiation sources located outside the field of view of an optical telescope; and (3) imaging target stray light in the field of view, which can be understood as light rays from targets that reach the focal plane of the detector via abnormal means.

Regarding the internal radiation stray light, its wavelengths are mostly distributed on the micrometer scale, so this kind of stray light mainly influences the infrared imaging system rather than the space-based wide-field surveillance systems applied to visible light in this paper. Hence, we ignore its effect on surveillance images. Non-target stray light beyond the field of view exists widely in all kinds of optical telescopes, especially in the

large field optical telescopes used to detect faint targets. After strong stray light enters the optical system, it causes a nonuniform noise signal to reach the detector, increasing the image gray values and spreading the gray scale from one edge to the other. The main reason for this is a gradual change in the material scattering intensity. Furthermore, since stray light is beyond the field of view, the gray value is maximized at the corresponding edge of a surveillance image, as shown in Figure 2a. We name this the first type of stray light background. Imaging target stray light in the field of view also causes a nonuniform noise signal and increases the gray values of surveillance images, resulting in a relatively bright area in the surveillance image with gray scale spreading from the center to the periphery, as shown in Figure 2b. The primary cause of this is the complete reflection in the lens that occurs when light from brighter stars reaches the lens at a specific angle. We name this the second type of stray light background. Any complex image can be regarded as a combination of these two forms of background.

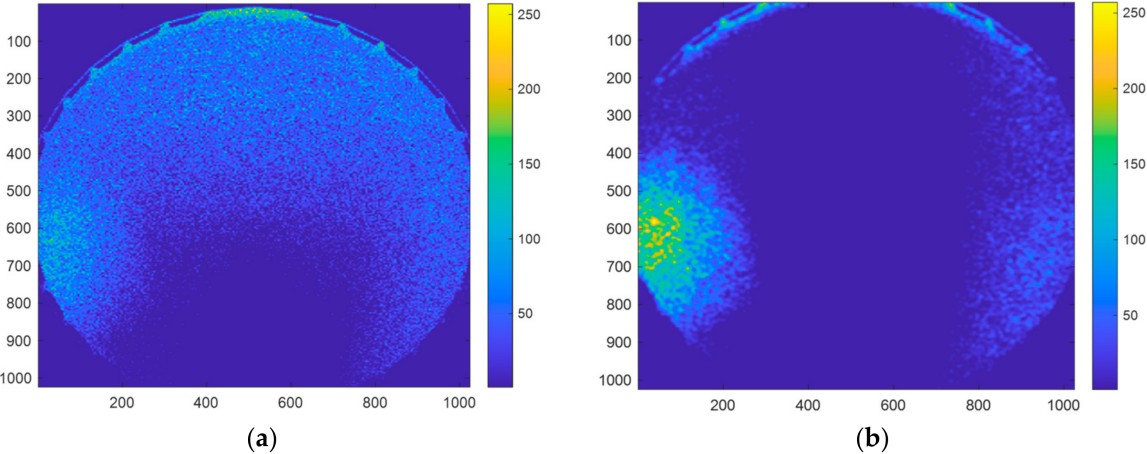

**Figure 2.** Two types of simulated stray light background gray-scale images. (**a**) the first type; (**b**) the second type.

To better comprehend the features of nonuniform backgrounds in surveillance images, we will analyze them using energy transfer equations [26]. To simplify the complex stray light transmission process, we can divide the transmission path into several parts, including several emitting and receiving surfaces where the receiving surface of each process is the emitting surface of the next. The transmission of stray light on any two surfaces conforms to the radiation transfer theory [27], the principle of which is shown in Figure 3.

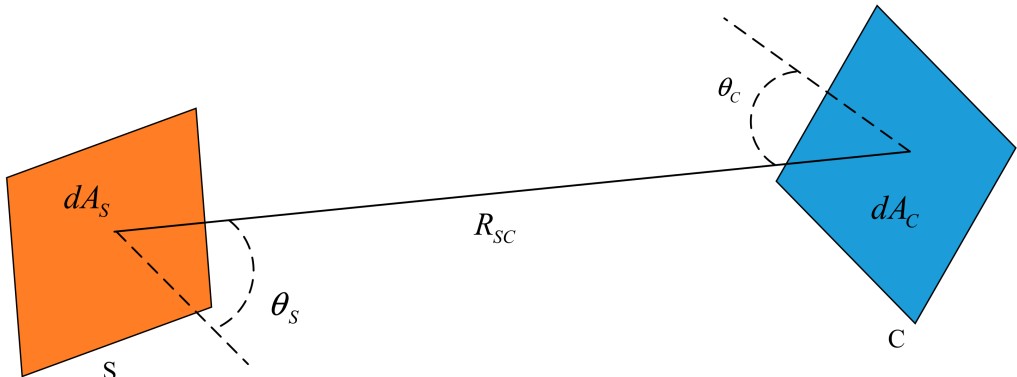

**Figure 3.** Radiation transfer schematic.

According to the radiative transfer theory, between two media surfaces, light energy propagates as follows [28]:

$$\mathrm{d}\Phi_C = \frac{L_S \mathrm{d}A_S \mathrm{d}A_C \cos\theta_C \cos\theta_S}{R_{SC}{}^2} \tag{1}$$

where $A_C$ and $A_S$ are the areas of the receiving and the source surfaces, respectively, $R_{SC}$ refers to the center length between the source and receiving surfaces, $L_s$ is the radiance of the source surface, $\Phi_C$ represents the receiving surface flux, and $\theta_C$ and $\theta_S$ are the angles between the center line and the normal line of the respective surfaces. Equation (1) can be simplified by breaking it into three parts, as follows [29]:

$$\mathrm{d}\Phi_C = \left(\frac{L_S}{E_S}\right)(E_S \cdot \mathrm{d}A_S)\left(\frac{\cos\theta_S \cdot \cos\theta_C \cdot \mathrm{d}A_C}{R_{SC}{}^2}\right) \tag{2}$$

$$\mathrm{d}\Phi_C = \mathrm{BRDF} \cdot \mathrm{d}\Phi_S \cdot \mathrm{d}\Omega_{\mathrm{sc}} \tag{3}$$

where $E_S$ represents the incident irradiance, $\mathrm{d}\Phi_S$ is the output flux, and BRDF is the bidirectional reflectance distribution function, which refers to the scattering properties of the material surface and defines the ratio of the scattering radiance to the incident irradiance of a rough surface, and $\mathrm{d}\Omega_{SC}$ represents the projected solid angle between the source and receiving surfaces. In terms of the differential form of Equations (2) and (3), we are roughly aware that the distribution of a nonuniform background affected by stray light exhibits a gradual form rather than an abrupt change.

In general, for space-based wide-field surveillance systems, the influence of stray light can be suppressed by optomechanical structures such as the hood and light-blocking ring [30]. However, the absorption rates of baffles and blades are always between 95% and 97%. At this time, the rest of the stray light will still be received by the detector, resulting in a nonuniform background. We still need to eliminate the background through image processing.

The surveillance image is described in the following way:

$$F(i,j) = T(i,j) + S(i,j) + B(i,j) + N(i,j) \tag{4}$$

where $(i,j)$ denotes the pixel coordinates, $F$ is an original surveillance image, $T$ refers to the space target, $S$ is the star, the light from stars is the cause of the second type of stray light background, and $B$ represents the background and $N$ refers to the noise. The fundamental principle of nonuniform background elimination is to estimate $B$ accurately and robustly while preserving $S$ and $T$. Based on the above analysis, an original surveillance image $F$ has the following two features: (1) Most of the pixels in $F$ are occupied by background-region pixels, whose gray values are different from those of target-region pixels, and (2) the nonuniform background caused by stray light exhibits a gradual rather than an abrupt change. Based on the first feature of the original surveillance image, we define two structural operators to take advantage of this difference. Based on the second feature, a morphology-based method can be used to eliminate the nonuniform background, since it features a slow change.

## 3. Nonuniform Background Elimination

In this section, we will introduce the IBSGM method in detail for eliminating nonuniform background while retaining stars and space targets.

### 3.1. Definition of Structural Operators

Based on the features of surveillance images described in Section 2, we can eliminate stray light nonuniform backgrounds by using the information of varying gray values between the target and background pixels. We define two structural operators, $\Delta B$ and $B_b$, to better utilize this difference, as shown in Figure 4a,b.

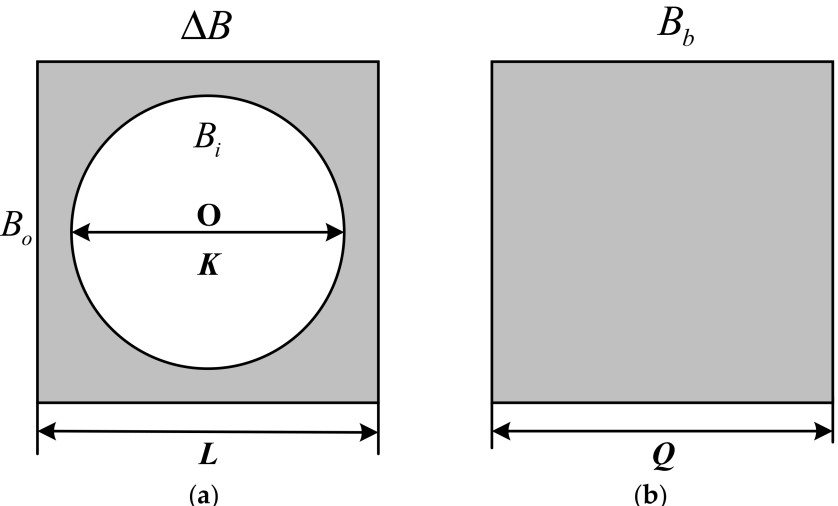

**Figure 4.** Defined structural operators used in the IBGSM method. (**a**) structural operator $\Delta B$; (**b**) structural operator $B_b$.

In Figure 4, $B_o$ and $B_i$ are defined as the outer and inner structural operators of $\Delta B$, respectively. $\Delta B = B_o - B_i$ refers to the ring-like area between $B_o$ and $B_i$. $B_b$ represents a uniform square area, whose size is bigger than $B_o$, and $K$, $L$ and $Q$ refer to the sizes of $B_i$, $B_o$ and $B_b$, respectively, with $K < L < Q$.

For the structural operator $\Delta B$, the coordinate of its center point $O$ is $(x_0, y_0)$, $x \in \left(-\frac{L}{2}, \frac{L}{2}\right)$, $y \in \left(-\frac{L}{2}, \frac{L}{2}\right)$.

$$\Delta B = f(x,y) = \begin{cases} 1, & \sqrt{(x-x_0)^2 + (y-y_0)^2} > \frac{K}{2} \\ 0, & \sqrt{(x-x_0)^2 + (y-y_0)^2} \le \frac{K}{2} \end{cases} \tag{5}$$

For the structural operator $B_b$, $x \in \left(-\frac{Q}{2}, \frac{Q}{2}\right)$, $y \in \left(-\frac{Q}{2}, \frac{Q}{2}\right)$.

$$B_b = g(x,y) = 1 \tag{6}$$

With the current algorithms, due to all the pixels covered in the filter being used for calculation, it is difficult to distinguish a complex background region from a target region with high accuracy. The structural operators that we define contribute to resolving it.

### 3.2. Self-Adaptive Size Adjustment

Surveillance images contain targets of different sizes. A structural operator with a fixed size will influence the background elimination effect and even reduce the information of faint targets. Before performing morphological processing, we block the original surveillance image and find the optimal size of the structural operator suitable for each image block by the minimum bounding rectangle method. To solve the problem of surveillance images being sensitive to the fixed-size structural operators used in other methods, the self-adaptive size adjustment of the structural operator is conducted as follows:

An original surveillance image $F_0$ with $1024 \times 1024$ imaging pixels is divided into several image blocks $f_c$ with $32 \times 32$ pixels, where c is the index of the image blocks. Each block is binarized with the threshold obtained by the following equations:

$$Th_c = \mu_c + \alpha\sigma_c \quad (1 \le c \le k) \tag{7}$$

$$\mu_c = \frac{1}{M \times N}\sum_{i=1}^{M}\sum_{j=1}^{N} f(i,j) \tag{8}$$

$$\sigma_c = \sqrt{\frac{1}{M \times N} \sum_{i=1}^{M} \sum_{j=1}^{N} (f(i,j) - \mu_c)^2} \tag{9}$$

where $k$ represents the number of image blocks, $\mu_c$ and $\sigma_c$ are the mean and standard deviation of each block, respectively, $M$ and $N$ refer to the rows and columns of each image block, respectively, and $\alpha$ is a coefficient, where $\alpha = 2$ is selected according to the features of the surveillance images.

The binarized image $b_c$ can be obtained by:

$$b_c(i,j) = \begin{cases} 0, & f(i,j) < Th_c \\ 1, & f(i,j) \geq Th_c \end{cases} \tag{10}$$

In order to determine the size ($K$ and $L$) of the optimal structural operator $\Delta B$ for each image block $f_c$, we find the set of the minimum bounding rectangle sides of the connected domain in each binary image block $b_c$. As shown in Figure 5, the value of the largest side in each set is used as the $K$ value of the structural operator $\Delta B$, while the value of parameter $L$ can be obtained by the value of $K$. Since the sizes of the largest targets differ in different image blocks, we use self-adaptive adjustable structural operators to perform morphological operations.

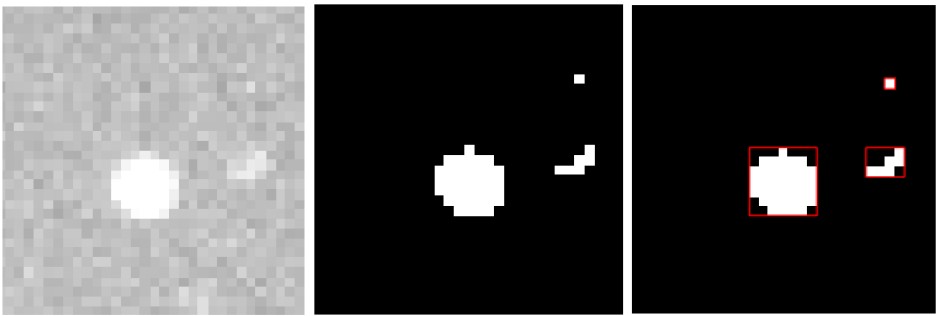

**Figure 5.** Process of finding the maximum target size in an image block.

### 3.3. Gray-Scale Morphological Operation

First, we use the self-adaptive adjustable structural operator $\Delta B$ to execute a dilation operation $\oplus$ on each image block $f_c$, and then stitch the operated image blocks into the size of the original image to obtain image $F_1$, as shown in Equation (11).

$$F_1 = \sum_{c=1}^{k} f_c(i,j) \oplus \Delta B = \sum_{c=1}^{k} \max \left\{ f_c(i-m, j-n) + \Delta B(m,n) \Big| (i-m),(j-n) \in D_{f_c}, (m,n) \in D_{\Delta B} \right\} \tag{11}$$

where $D_{f_c}$ and $D_{\Delta B}$ refer to the domains of $f_c$ and $\Delta B$, respectively. In terms of the definition of the dilation operation, if a target is exactly in the internal structural operator $B_i$, since it is not in the defined domain of $\Delta B$, the pixel value of this part will not participate in the dilation procedure. As a result, the dilation process will only use pixels in the background surrounding targets. In this way, the target pixels are replaced by background pixels to protect them from being eliminated. We choose the value of the largest side in each image block to be the $K$-value of the structural operator $\Delta B$ to ensure that targets are in the internal structural operator $B_i$. As most stars and space targets are circular, the internal structural operator $B_i$, which is also circular, can perform the replacement most accurately. This is equivalent to a "keying" operation and fundamentally improves the accuracy of nonuniform background elimination.

Regarding the two parameters of $\Delta B$, $K$ is ascertained in Section 3.2. For $L$, it controls the number of background pixels involved in calculation. If $L$ is too large, some targets will mistake their neighboring target pixels as their surrounding background pixels during the dilation operation. Incorrect replacement causes mutual interference between targets. Therefore, we assign $L$ a value of $K + 2$, which not only avoids interference, but also ensures

that enough background pixels are included in the calculation, thereby ensuring accuracy. The result of this process is shown in Figure 6b.

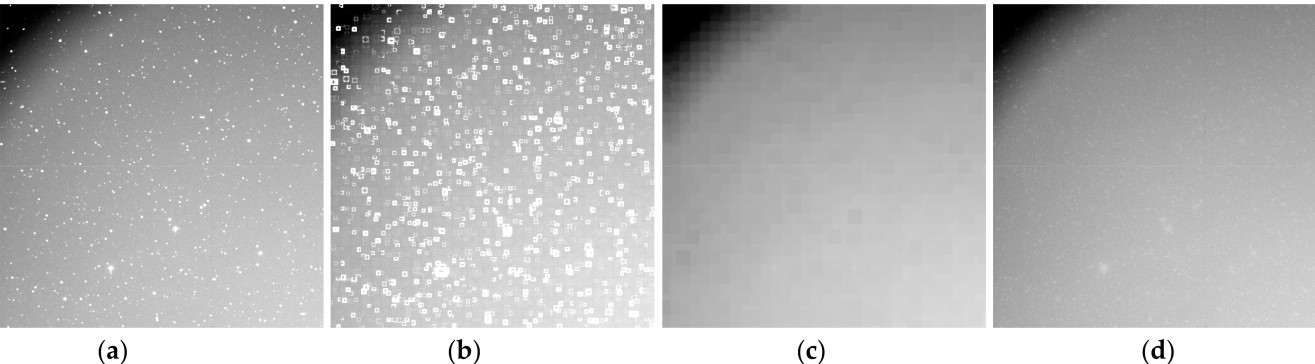

(**a**)　　　　　　　　　　　(**b**)　　　　　　　　　　　(**c**)　　　　　　　　　　　(**d**)

**Figure 6.** Morphological operation results. (**a**) Original real surveillance image; (**b**) dilation operation result after Equation (11); (**c**) erosion operation result after Equation (12); and (**d**) minimum value result after Equation (14).

Then, as described by Equation (12), we employ the structural operator $B_b$ to execute an erosion operation $\Theta$ on image $F_1$.

$$F_2 = F_1 \Theta B_b = \min\left\{F_1(i+m, j+n) - B_b(m,n) \big| (i+m), (j+n) \in D_{F_1}, (m,n) \in D_{B_b}\right\} \tag{12}$$

where $D_{F_1}$ and $D_{B_b}$ refer to the domains of $F_1$ and $B_b$, respectively. The optimal structural operator $\Delta B$ for each image block is presented in Section 3.2. If the target is too large (bigger than $32 \times 32$ pixels) or does not appear completely within an image block, some target pixels will be mistaken as background pixels and will remain in image $F_1$. Hence, we need to retrieve these lost targets by performing an erosion operation through the structural operator $B_b$ on image $F_1$. Considering the previously acquired surveillance image information, the majority of targets will not exceed $50 \times 50$ pixels. Furthermore, in terms of the surveillance image features described in Section 2, most of the pixels in $F$ are occupied by background-region pixels. As a result, we set the value of parameter $Q$ to 50, which not only ensures that large targets are retrieved, but also confirms that stray light backgrounds are not confused for targets.

Moreover, the dilation operation in Equation (11) will improve the overall gray value of the image and dilate the range of the nonuniform background region. The erosion operation not only adjusts the overall brightness of image $F_1$, but also further reduces the target-region gray value that is replaced by the surrounding background region in the dilation process, which ensures that the brightness of the target is not overly diminished. The result is shown in Figure 6c.

Since the size of $B_b$ in the erosion process is larger than the size of $\Delta B$ in the dilation process $(Q > L)$, it reduces the nonuniform background region that needs to be eliminated. Consequently, we reuse $B_b$ to execute a dilation operation on image $F_2$ as shown in Equation (13). In short, the nonuniform background region that needs to be eliminated will not change.

$$F_3 = F_2 \oplus B_b = \max\left\{F_2(i-m, j-n) + B_b(m,n) \big| (i-m), (j-n) \in D_{F_2}, (m,n) \in D_{B_b}\right\} \tag{13}$$

The above is the case when targets are fully or partially in the definition domain of the inner structural operator $B_i$. However, if there are no target pixels in the definition domain of $B_i$, at this time, the relationship of this substitution is uncertain. Therefore, we take the minimum value of $F_3$ and the original image $F_0$ as shown in Equation (14).

$$F_4 = \min(F_0, F_3) \tag{14}$$

The result is shown in Figure 6d, where $F_4$ is the final nonuniform background needing to be eliminated.

Based on the IBSGM method, we can eliminate nonuniform backgrounds accurately while retaining stars and space targets.

## 4. Experiments and Discussion

To verify the advantages of the IBSGM method, we compared it with three other methods: top-hat transformation (THT), mean iterative filtering (MIF), improved new top-hat transformation (INTHT). These methods were used with the same simulated and real images.

### 4.1. Simulation Experimental Principles and Results

A flowchart of the simulation image experiment is shown in Figure 7.

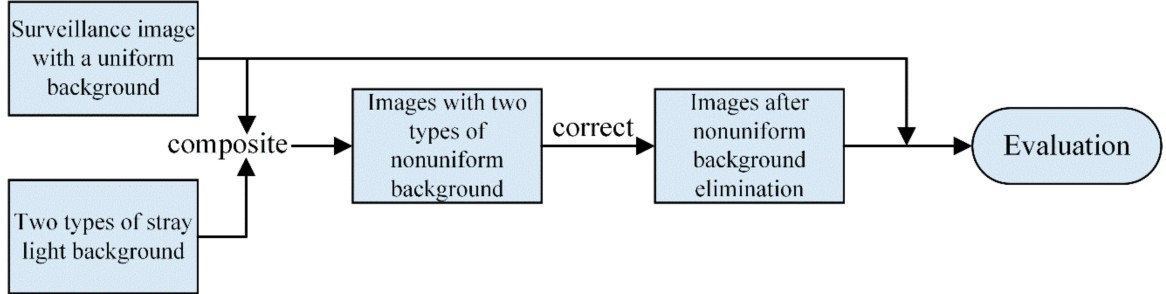

**Figure 7.** Flowchart of the simulation image experiment.

First, an image only containing stars and space targets with a uniform background was simulated and generated according to the Tycho-2 Catalogue and the relevant parameters of the optical system with a 80° × 35° field of view. Tycho-2 Catalogue contains a large number of stars and has a limiting magnitude V~11 and mean sky density of 60 objects per square degree. It was used as a surveillance image without a nonuniform background, as shown in Figure 8a. In the simulation process, the parameter of the GSENSE6060 detector produced by Gpixel incorporation were referenced, which has a 10 μm pixel size and 3 s exposure time. Typical PSF (point spread function) could be fitted to a gaussian function. Then, we simulated the two types of stray light backgrounds described in Section 2 by ray tracing, as shown in Figure 2, which are synthesized with the simulated uniform surveillance image. Two types of simulation surveillance images with nonuniform backgrounds are shown in Figure 8b,c. Finally, different algorithms are used to eliminate the two types of stray light nonuniform backgrounds in the simulated surveillance images. The experimental results are shown in Figures 9 and 10.

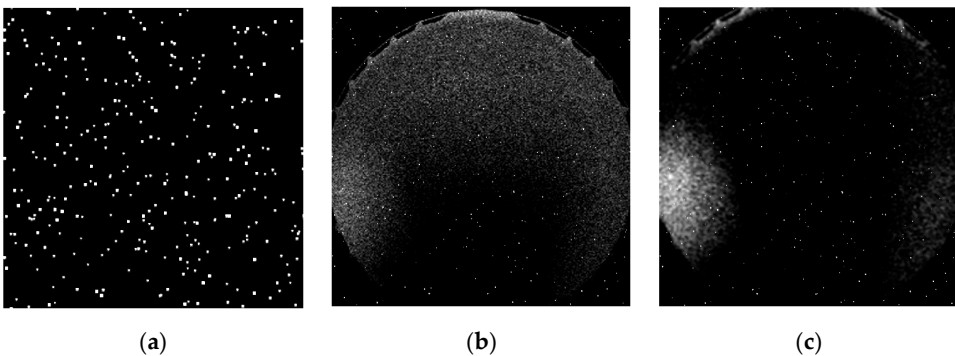

(a)　　　　　　　　　　　　(b)　　　　　　　　　　　　(c)

**Figure 8.** Simulated surveillance images with (**a**) a uniform background; (**b**) the first type of stray light background; and (**c**) the second type of stray light background.

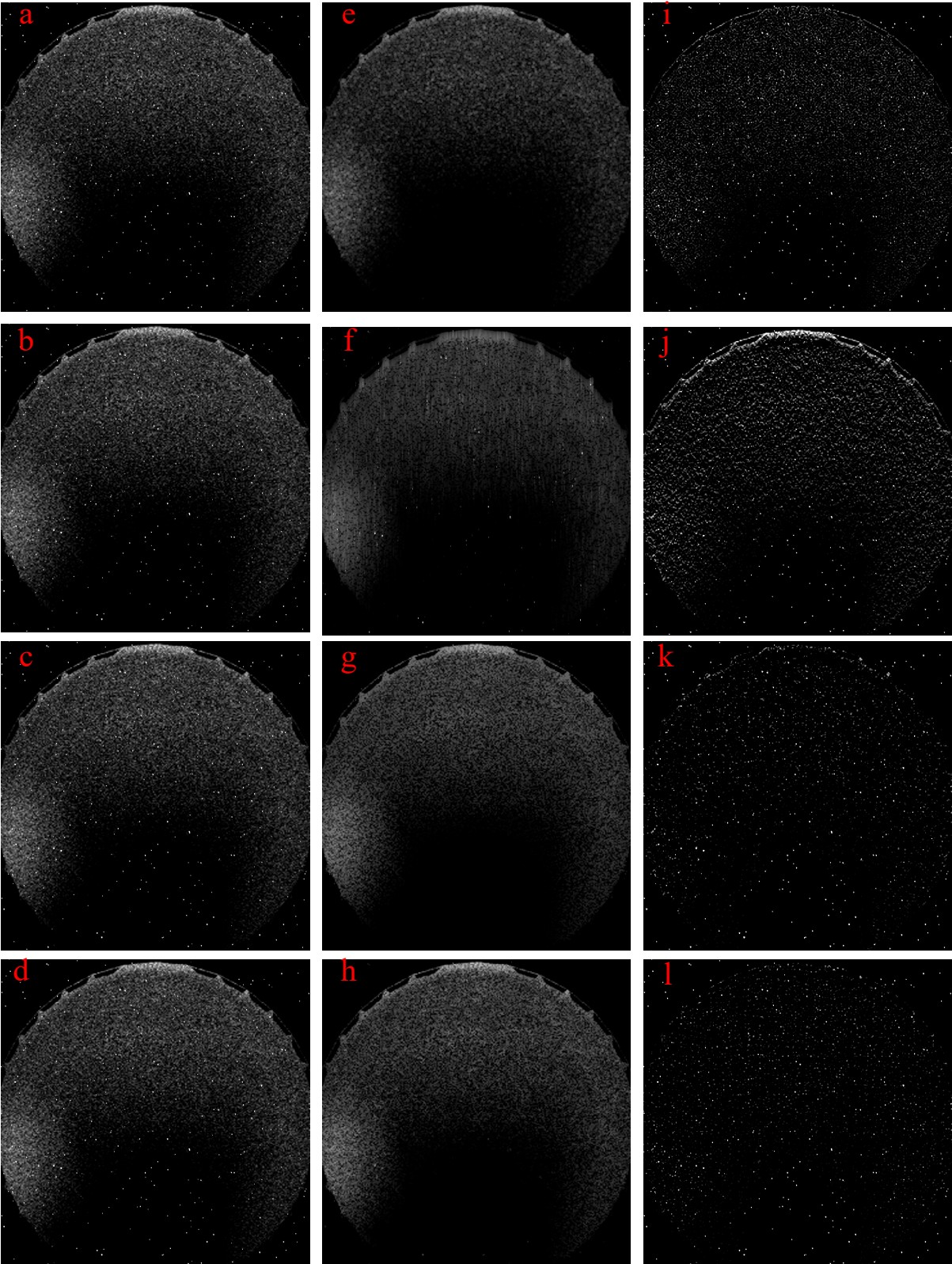

**Figure 9.** Elimination result for the first type of stray light background. (**a–d**) Simulated original surveillance images; (**e–h**) nonuniform backgrounds estimated by the THT, MIF, INTHT and IBSGM methods, respectively; and (**i–l**) background elimination results of the THT, MIF, INTHT, and IBSGM methods, respectively.

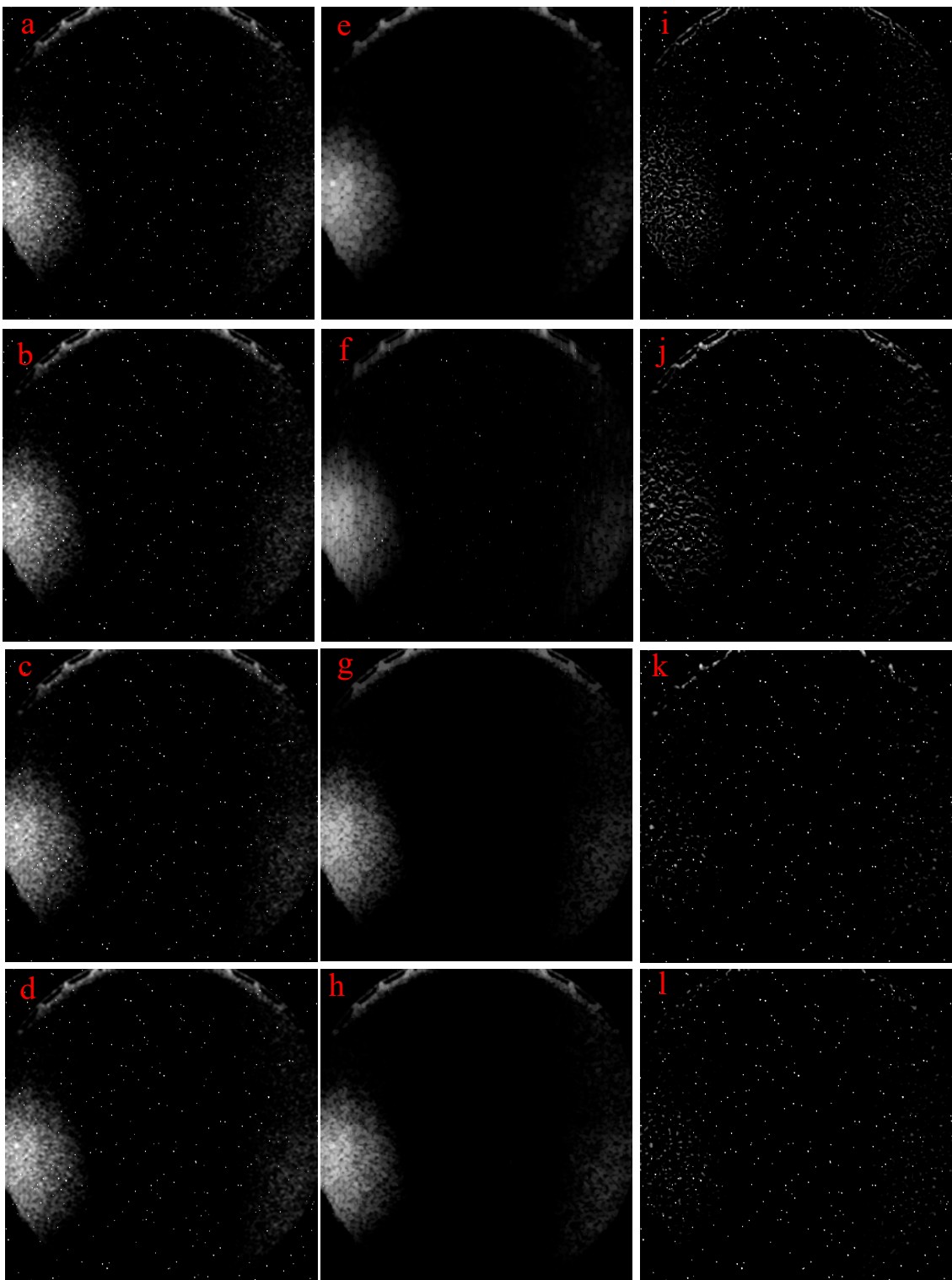

**Figure 10.** Elimination result for the second type of stray light background. (**a**–**d**) Simulated original surveillance images; (**e**–**h**) nonuniform backgrounds estimated by the THT, MIF, INTHT, and IBSGM methods, respectively; and (**i**–**l**) background elimination results of the THT, MIF, INTHT, and IBSGM methods, respectively.

For the evaluation of the reference image, we use the root mean square error (RMSE) to evaluate the effect of nonuniform background elimination. Here, $O$ and $I$ are the simulated

surveillance images with a uniform background and the processed surveillance image, respectively.

$$RMSE = \sqrt{\frac{1}{m \times n} \sum_{i=1}^{m} \sum_{j=1}^{n} (O(i,j) - I(i,j))^2} \tag{15}$$

where $m$ and $n$ refer to the rows and columns of $O$ and $I$, respectively.

The smaller the RMSE value, the closer the corrected image is to the original image with a uniform background, and the better the ability of the algorithm to eliminate the stray light nonuniform background. The RMSE results are shown in Table 1.

**Table 1.** RMSEs of the surveillance images by different algorithms.

| Stray Light Background | THT | MIF | INTHT | IBSGM |
|---|---|---|---|---|
| First type | 8.4982 | 12.4052 | 5.0044 | 4.8320 |
| Second type | 4.7121 | 7.2477 | 3.1371 | 2.6711 |

Combining the results of Figures 9 and 10 and Table 1, the proposed IBSGM method has higher accuracy in eliminating both types of stray light nonuniform backgrounds. Therefore, the image processed image by our method is closer to the original simulated image with a uniform background. Since more complex situations cannot be studied by simulation, we used real images to further verify the performance of IBSGM and analyze its advantages in comparison with other methods in the next subsection.

*4.2. Real Image Experimental Results and Discussion*

Real surveillance images were captured by a CMOS (complementary metal-oxide-semiconductor) sensor with a 3 s exposure time. The detector has 10 K × 10 K pixels, $10° × 10°$ field of view and a 12-bit gray level. To see the experimental results more intuitively, we cropped the image to 1024 × 1024 pixels. The results of nonuniform background elimination are shown in Figure 11, where (a–c) are original surveillance images and (d–o) refer to those obtained after eliminating the nonuniform backgrounds in various ways.

Unlike the simulation experiment, in the real image experiment, we cannot obtain images without nonuniform backgrounds that only contain stars and space targets. As a result, we cannot evaluate the effect of nonuniform background elimination in terms of RMSE.

4.2.1. Accuracy of Nonuniform Background Elimination

To quantitatively compare the accuracy of the above four methods, we adopt residual analysis. In the residual image, since the majority of pixels are background pixels, the greater the mean, the higher the image's overall gray value, so the residual background of the first type of stray light is greater. The greater the standard deviation, the higher the degree of gray-value fluctuation, so that more of the second type of stray light background remains. In a word, the smaller the mean and standard deviation, the higher the accuracy of the correction algorithm.

In the residual image, we applied an exclusion domain to remove the interference of targets before calculating the mean and standard deviation. Specifically, the threshold $Th$ is obtained by adaptive threshold segmentation. We then establish the exclusion domain $e_B$ as:

$$e_B(i,j) = \begin{cases} 0, & f_B(i,j) \geq Th \\ 1, & f_B(i,j) < Th \end{cases} \tag{16}$$

where $f_B$ refers to the processed surveillance image. Finally, we obtain the residual image $R$ as shown in Equation (17):

$$R(i,j) = f_B(i,j)e_B(i,j) \tag{17}$$

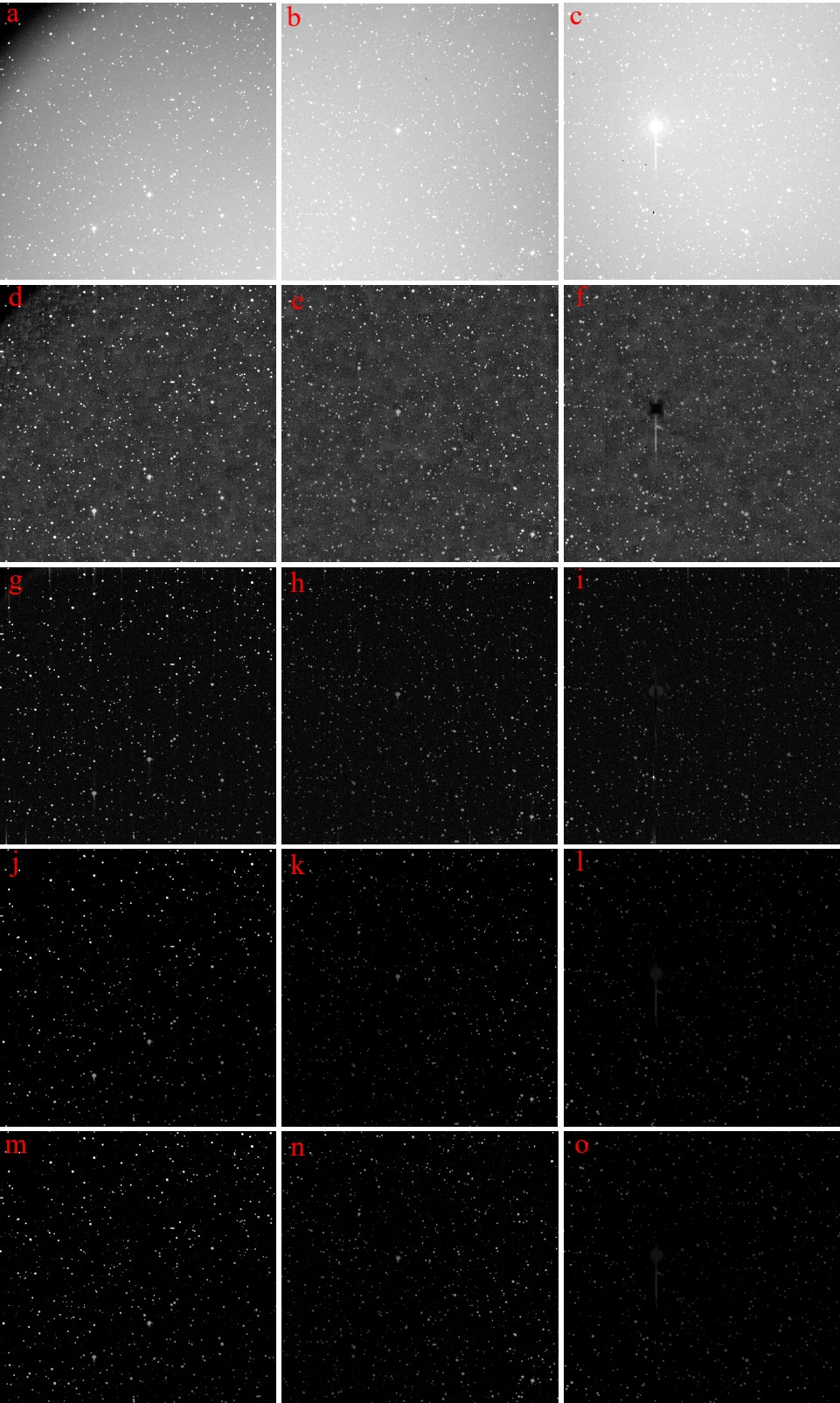

**Figure 11.** Nonuniform background elimination results. (**a–c**) Original real surveillance images; background elimination results of (**d–f**) THT; (**g–i**) MIF; (**j–l**) INTHT; and (**m–o**) IBSGM.

The results of the residual image means and standard deviations obtained by different algorithms are shown in Table 2.

**Table 2.** Means and standard deviations of the residual image obtained by different methods.

| Background Residual | Figure 11a | | Figure 11b | | Figure 11c | |
|---|---|---|---|---|---|---|
| | Mean | Standard Deviation | Mean | Standard Deviation | Mean | Standard Deviation |
| THT | 21.2518 | 9.3417 | 21.7236 | 9.1999 | 21.8202 | 9.3119 |
| MIF | 3.1559 | 3.0220 | 2.1904 | 2.4297 | 2.4088 | 2.5845 |
| INTHT | 0.4408 | 0.4983 | 0.4270 | 0.4077 | 0.5071 | 0.5887 |
| IBSGM | 0.0231 | 0.1502 | 0.0249 | 0.1559 | 0.0245 | 0.1736 |

In the THT method, all pixels (target and background regions) covered by the fixed structural operator participate in the calculation to eliminate the background, which leaves a great deal of nonuniform background noise in the residual image. Therefore, the accuracy of THT is relatively low. In the MIF method (five iterations), as in the THT method, the size of the filter is fixed and the pixels (including target and background) participate in the calculation, making it difficult to further improve the accuracy in complex environments even by increasing the number of iterations. The INTHT method, in contrast to the above two methods, uses structural operators with different domains to decrease the target regions involved in the calculation. However, it uses a fixed-size structural operator to distinguish target and background pixels, which does not work well for smaller-sized targets and does not consider the sensitivity of the structural operator. The IBSGM method defines two structural operators with a size that constantly self-adapts to the size of the target. The optimum structural operators are used to perform morphological operations in each image block. Thereby, the accuracy of nonuniform background elimination is greatly improved. We can see that IBSGM has substantially greater accuracy in eliminating stray light nonuniform backgrounds compared with the other methods.

### 4.2.2. Accuracy of Target Retention

The purpose of nonuniform background elimination is to better ensure subsequent target recognition. Therefore, the algorithm should preserve targets in surveillance images as much as possible. To see the result more intuitively, a nonuniform background requiring elimination is shown in Figure 12, where (a–c) are the original surveillance images and (d-o) refer to the nonuniform background that needs to be estimated and eliminated by different algorithms.

To gain a better idea of how accurate different algorithms are at retaining their targets, we determined the number of targets that are retained after background elimination based on a connected domain method. The target retention results are shown in Table 3.

**Table 3.** Comparison of target retention rates using different methods.

| Method | Figure 12a | Figure 12b | Figure 12c |
|---|---|---|---|
| THT | 83% | 85% | 81% |
| MIF | 86% | 88% | 85% |
| INTHT | 93% | 96% | 92% |
| IBSGM | 98% | 99% | 97% |

In the THT method, as explained in Section 4.2.1, the accuracy of nonuniform background elimination is relatively low. This means that some nonuniform background noise remains in the processed image, significantly reducing target detection accuracy. In the MIF method (5 iterations), the target gray value is decreased to some extent after several iterations, resulting in the loss of low-SNR targets. Moreover, certain brighter targets are mistaken for the nonuniform background noise that is to be deleted. The INTHT method is limited by the fixed-size structural operator, leading to some targets being mistaken for the background and lost. Hence, its target retention accuracy still needs to be improved.

Meanwhile, with the IBSGM method, two structural operators are constructed with a size that constantly self-adapts to the target size and combined with morphological operations. This eliminates nonuniform backgrounds with high precision and robustness while retaining more targets. In some instances, the rate of target retention cannot reach 100% due to interference from too many targets in the surveillance image.

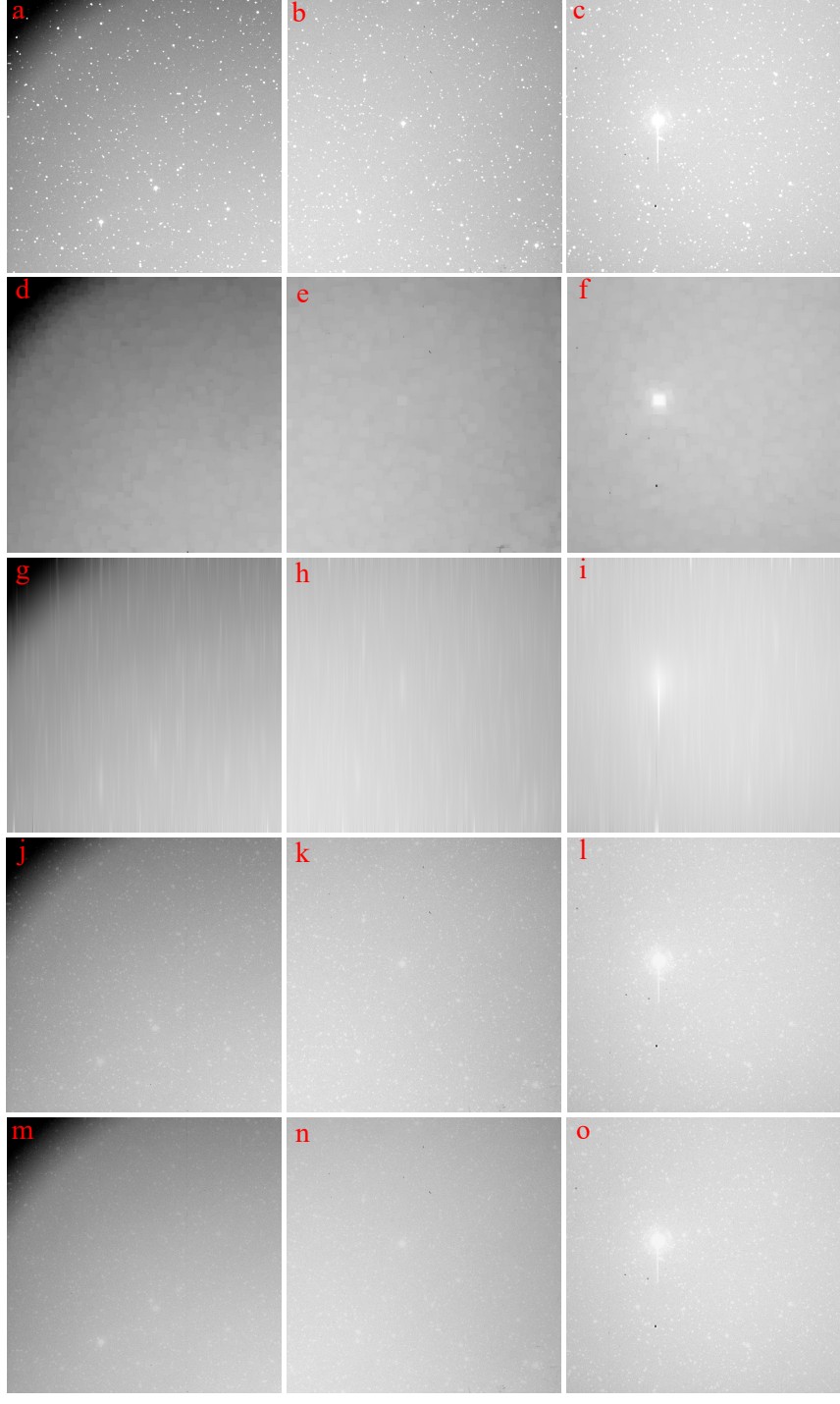

**Figure 12.** Nonuniform background estimation by different algorithms. (**a**–**c**) Real original surveillance images. Nonuniform backgrounds estimated by (**d**–**f**) THT; (**g**–**i**) MIF; (**j**–**l**) INTHT; and (**m**–**o**) IBSGM.

### 4.2.3. Computation Time

Table 4 compares the computation times taken for different methods to process a $1024 \times 1024$ test image. Image processing was undertaken in MATLAB R2020b on a PC computer with an i5-9400f CPU (2.90 GHz) and 16 GB of main memory.

**Table 4.** Comparison of the computation times of different methods.

| Method | Computation Time (s) |
|--------|---------------------|
| THT | 0.517 |
| MIF | 3.521 |
| INTHT | 0.436 |
| IBSGM | 6.934 |

Although the computation time of the proposed IBSGM method was higher due to the self-adaptation process, the corresponding accuracy of nonuniform background elimination and target retention were greatly improved, which is exactly what is required for target recognition.

### 5. Conclusions

Stray light nonuniform background elimination is not only a requirement of space-based surveillance, but is also an essential prerequisite for subsequent target detection and tracking. To overcome the insufficiency of current methods in accurate stray light nonuniform background elimination, we proposed a robust and accurate elimination method based on image block self-adaptive gray-scale morphology (IBSGM).

In this study, we first analyzed the formation and elimination principles of stray light nonuniform backgrounds. Then, we defined two structural operators with different sizes and domains, which can make full use of the difference information between the target region and surrounding background region. Finally, we blocked the original surveillance image to obtain the optimal sizes of the structural operators suitable for performing morphological operations on each image block. The experimental results on simulated and real image datasets show that, compared with other methods, IBSGM has higher precision in eliminating nonuniform backgrounds with nearly no target losses.

**Author Contributions:** Conceptualization, J.W. and X.W.; methodology, J.W.; software, J.W.; validation, J.W. and Y.L.; formal analysis, J.W.; data curation, J.W.; writing—original draft preparation, J.W.; writing—review and editing, J.W. and Y.L.; supervision, X.W. and Y.L. All authors have read and agreed to the published version of the manuscript.

**Funding:** This research was funded by the Strategic Priority Research Program of Chinese Academy of Sciences, grant number XDA17010205.

**Institutional Review Board Statement:** Not applicable.

**Informed Consent Statement:** Not applicable.

**Data Availability Statement:** The study did not report any data.

**Acknowledgments:** The authors are grateful for the anonymous reviewers' critical comments and constructive suggestions.

**Conflicts of Interest:** The authors declare no conflict of interest.

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
