# Peer review of "Stray Light Nonuniform Background Elimination Method Based on Image Block Self-Adaptive Gray-Scale Morphology for Wide-Field Surveillance"

_applsci, doi:10.3390/app12147299_

Round 1

Reviewer 1 Report

Please find my (partial) revision in the attachment.

Reviewer 2 Report

The stray light removal methodology proposed by the authors looks very effective and adopts a solid mathematical approach.
However, as an astronomer, I find that the typical images analysis tools we adopt should be at least tested on their images and the results compared. It is known that disentangling stray light and the detector response disuniformity could be difficult as the first represents an additive effect whereas the second is multiplicative, but in some cases it could be not crucial. It depends on the aim one wants to achieve. Of course in some cases stray light could cause a severe image degradation and a methodology like that presented here could help, but in other cases (e.g. when a precise estimate of the source magnitude/flux is not that important) a rough image clean (or none) could be sufficient.
There are several packages that can be tested and the authors are free to use any, but I would suggest Source Extractor (http://www.astromatic.net/software/sextractor) or its python port "sep" (https://github.com/kbarbary/sep). This code can also produce the objects' pixel mask (called "binarized image" by the authors). I consider this test useful and the results could complement those reported in the manuscript.

That said, I list here the major points that the authors should address/clarify before the paper can be published:

- The used images (simulated and real) seem to refer to a particular camera/instrument. That is OK but the reader should be informed of the basic characteristics: pixel size, typical PSF, size / field of view (are they all the same?), limiting magnitude/flux reached in 1-s (or N-s).

- The meaning of "gray-scale" is clearly referred to "intensity" or flux, but sometimes it seems to refer to the background or "baseline" level of the image. To avoid misunderstanding, I would suggest making its usage more consistent within the manuscript.

- The actual number of "satellites in orbit" is an ambiguous concept. Space-Track (https://www.space-track.org/) reports 8948 "payloads" orbiting Earth (July 8th 2022). To these one must add some 2300 rocket body parts + other stuff.
  Celestrak lists 9118 payloads on orbit (https://celestrak.org/satcat/boxscore.php). And of course many more are the debris.
  It would be nice to get some statistical evaluation on the potential number of objects targeted by a camera like that considered by the authors. In fact the mentioned Surveillance Systems are more relevant to detect and track the location of objects whose orbit is not known well (debris are an example) and eventually to know their origin and characteristics, and predict what they will be doing in the future.

- On page 6, Fig. 4, the relative sizes of the "structural operators" are defined, but their quantification is only reported on page 8 (L=K+2 and Q=50). That looks again as something specific to the camera considered by the authors. It could be fine, but it should be stated what criteria apply for a generic camera/detector, based on its characteristics.

- The authors should quote at least the mean objects' density in their images. They mention the Tycho-2 catalogue, which has a limiting magnitude V ~ 11 and mean sky density of 60 objects per square degree. Are these reference figures also for their images?

Other minor points:
- Figure 1 and 7, reporting flowcharts, should be in vector format (e.g. PDF).

- Images in Figure 2 do not have the correct size.

- Some sentences are quite confusing and should be rephrased. For example the paragraphs at rows 265-268 and 382-386. There are quite a few typos and punctuation errors too. I suggest a careful review of the text.

- I assume the used data (simulated and real) are not privately available, so I would make them accessible to the reader.
  Even if the software implementation of the method seems straightforward, it would be appreciated if the authors make it available, at least in a simplified or precompiled form.

- It would be nice to report some figures for the images processing time. I guess real-time processing is viable with typical hardware resources.

Reviewer 3 Report

In this paper authors proposed a robust and accurate nonuniform background elimination method based on image block self-adaptive gray-scale morphology (IBSGM). The introduction part well described the state of the art of this methodology and clearly presented the problem to be solved. Their results revealed that IBSGM method has a higher precision nonuniform background elimination effect when compared to other methods. The paper is very well organized and presents interesting novel results which deserve to be published after minor corrections. Some minor points:

1.     Figure 2 is blurred, small and low resolution, it should be corrected.

2.     For equations 1,2 and 3 some references should be added. The same for all equations. If they are no new it is suggested a reference.

3.     Figure 4 a and b should be of same size. The meaning of figure b is not so clear, please describe more details in the text.

4.     I cannot see the differences between figures 6c and 6d, please explain better.
